# Prediction of clinical trial enrollment rates

**Cameron Bieganek[1], Constantin Aliferis[1,2], Sisi Ma**[1,2]*

**1** Institute for Health Informatics, University of Minnesota, Minneapolis, MN, United States of America,
**2** Department of Medicine, University of Minnesota, Minneapolis, MN, United States of America

* sisima@umn.edu

## Abstract

Clinical trials represent a critical milestone of translational and clinical sciences. However, poor recruitment to clinical trials has been a long standing problem affecting institutions all over the world. One way to reduce the cost incurred by insufficient enrollment is to minimize initiating trials that are most likely to fall short of their enrollment goal. Hence, the ability to predict which proposed trials will meet enrollment goals prior to the start of the trial is highly beneficial. In the current study, we leveraged a data set extracted from *ClinicalTrials.gov* that consists of 46,724 U.S. based clinical trials from 1990 to 2020. We constructed 4,636 candidate predictors based on data collected by *ClinicalTrials.gov* and external sources for enrollment rate prediction using various state-of-the-art machine learning methods. Taking advantage of a nested time series cross-validation design, our models resulted in good predictive performance that is generalizable to future data and stable over time. Moreover, information content analysis revealed the study design related features to be the most informative feature type regarding enrollment. Compared to the performance of models built with all features, the performance of models built with study design related features is only marginally worse ($AUC = 0.78 \pm 0.03$ vs. $AUC = 0.76 \pm 0.02$). The results presented can form the basis for data-driven decision support systems to assess whether proposed clinical trials would likely meet their enrollment goal.

**Data Availability Statement:** The data used in this study can be downloaded from the following urls: https://clinicaltrials.gov/ct2/resources/download https://www2.census.gov/programs-surveys/popest/tables/2010-2018/state/totals/PEP_2018_

## Introduction

Clinical trials represent a critical milestone of translational and clinical science with the most direct impact potential for advancing healthcare related outcomes. Patient recruitment is a necessary condition of success for clinical trials. Under specific situations such as the broad impact of COVID-19, rapid and high volume enrollment for vaccine trials is pivotal to global public health. However, poor recruitment to clinical trials has been a long standing problem affecting institutions in the US and all over the world. Institute of Medicine reports cited 71% of phase III NCI approved trials and 40% or more of NCI sponsored trials closed without meeting their enrollment goals [1, 2]. This incurs very significant costs and wasted resource for the trial sponsors, scientists conducting the trials, and society at large. Stated differently, it creates numerous dead ends for investigators and the translational health sciences enterprise. The National Institutes of Health has recognized these problems and has made improvements in clinical trial enrollment a major focus in recent years. Many initiatives, including the clinical

PEPANNRES.zip https://www.natureindex.com/annual-tables/2018/institution/all/all/.

**Funding:** SM's time on this work is partially supported by Grant UL1TR002494.

**Competing interests:** The authors have declared that no competing interests exist.

trials transformation initiative and the trial innovation network by the national center for advancing translational sciences, have been established with improving clinical trial enrollment as one of their primary missions [3, 4].

Many barriers to effective enrollment has been identified, including geographic and socioeconomic access, resource constraint, perception of the patients, interest and bandwidth of physicians, among other barriers [4–6]. Approaches for increasing clinical trial enrollment has been proposed. For example, utilizing specialized task force with dedicated resources for trial enrollment can better coordinate resources among multiple simultaneous trials and improve efficiency for enrollment. Establishing centralized clinical trial recruitment management systems with linkage to the electronic health record can facilitate enrollment by identifying potentially eligible patients according to their existing health records. Lastly, recruiting through alternative channels such as social media has also shown promise [7–10].

Improving enrollment for existing trial is critical, but it is only one side of the story. The other key challenge is the cost incurred by launching trials that are unlikely to meet their enrollment goal. The current study aims to minimize this cost by building models to predict the enrollment rate prior to the start of the trial. Accurate enrollment rate prediction model provides valuable information for the cost benefit analysis for launching future clinical trials and is a first step towards a decision support system for clinical trials.

Prior literature on clinical trial enrollment prediction falls into two broad categories. The first category uses various types of parametric models for enrollment rate prediction [11–15]. This class of model estimates the number of patients would enroll based on estimated recruitment rate specified by the researcher or the trial enrollment agency. Such models, if successfully built, have in principle the capacity to provide the expected number of patients recruited in a given period of time, as well as confidence intervals around the expected number. Several of the estimation methods also aims to predict recruitment under complex trial settings such as multi-center trials [13, 15]. The main drawback of this class of models is that the accuracy of their prediction heavily relies on the estimated recruitment rate, a piece of information that is generally unavailable prior to the initiation of the trial. Therefore, this class of methods have limited utility for enrollment evaluation prior to the launch of clinical trials. The second category of studies in the existing literature examined factors associated with enrollment rate [7, 16–20]. Factors that have been previously reported to be associated with enrollment rate include recruitment strategy, trial design, seasonality, type of disease studied, disease severity of the potential participants, and their socioeconomic characteristics. However, the majority of the studies investigating the factors influencing enrollment rate only examined a small number of predictor variables, which may result in sub-optimal models for enrollment prediction. Further, the data used for these studies typically came from narrow trial populations. For example, the data are often obtained from single health systems, or restricted to specific diseases. As a result, these identified factors and their quantitative relationship with enrollment rate are less likely to generalize to the broad spectrum of clinical trials conducted nationally.

The main objective of the current study is to build models for clinical trial enrollment prediction using available information prior to the initiation of the trial. To address the above mentioned gaps in the literature, we specified four sub-objectives and briefly describe the strategies to achieve these sub-objectives: (1) To build generalizable models over different institutions and organizations that produce robust and stable performance over time: we leveraged a data set consisting of 46,724 U.S. based clinical trials from 1990 to 2020 extracted from *ClincalTrials.gov*, covering a broad trial population. (2) To examine a larger variety of features for enrollment prediction and improve predictive performance: we constructed a large number of candidate predictors regarding various aspects of individual clinical trials, including characteristics regarding the targeted trial population, the institution responsible for recruitment, the

medical domain of the trial, and the design of the trial. (3) To construct predictive models that do not rely on estimated enrollment rate and produce good performances: We applied a variety of state-of-the-art supervised learning methods to build predictive models for enrollment rate prediction. (4) To identify the feature type that is the most informative for enrollment rate prediction: we compared the information content for enrollment rate prediction among models constructed with different feature types.

The organization of the manuscript is as the following: In the methods section, we first describe the data acquisition and processing procedures. We then introduce the design of our analysis, including the time series cross-validation protocol for model selection and performance estimation, and the comparison of information content in various feature types. We also introduced the metrics for predictive performance examined. In the results section, we present the performance of various models for predicting enrollment rate. In the discussion section, we highlight the contribution of the study, note the limitations, and point to potential future work.

The main contribution of the current study is that we constructed generalizable models for clinical trials enrollment prediction with good predictive performance and are stable over time. Our results empirically demonstrated the feasibility of predicting enrollment rate prior to the initiation of clinical trials. Thus, this study forms the basis for data-driven decision support systems to assess whether proposed clinical trials would likely meet their enrollment goal.

## Methods

### Data

Our primary data source is *ClinicalTrials.gov*, a Web-based resource with information on publicly and privately supported clinical studies on a wide range of diseases and conditions, maintained by the National Library of Medicine (NLM). The information regarding a specific clinical study is provided and updated by the sponsor or principal investigator of the clinical study and available for download.

We downloaded the XML formatted clinical trial records for all studies on the website on October 28, 2020 from https://clinicaltrials.gov/ct2/resources/download. We restricted our dataset to completed, U.S. based, interventional clinical studies (i.e. clinical trials). The *ClinicalTrials.gov* website contains data on over 350,000 clinical trials conducted around the world over the last two decades. Filtering this data set to completed, U.S. based clinical trials reduced the number of studies to around 56,000. We implemented an additional filtering step in order to only include studies where the enrollment was listed as "Actual" rather than "Estimated" and to only include studies with a duration of at least 5 days. The final data set included 46,724 studies. The total number of trials by year is shown in Fig 1.

**Outcome of interest.**   The target variable of interest for this study was clinical trial enrollment rate. The rate is defined as the total enrollment divided by the study duration, where the total enrollment and study duration were extracted from the XML records. A box plot of the enrollment rates for each year is shown in S2 Fig in S1 File. For enrollment rate prediction, clinical trial enrollment rate, $r$ (number of participants per year), was categorized into three levels: *low*, *medium*, and *high*. The *low* category was defined to be $r \leq 25$, the *medium* category was defined to be $25 < r \leq 100$, and the *high* category was defined to be $r > 100$. Enrollment rates of 25 and 100 correspond to the 48th and 77th percentiles of the enrollment rate distribution. There is a time-dependent drift in the distribution of class membership, which can be seen in Fig 2. Some of the drift in the more recent years is due to a selection bias, which is discussed in a later section.

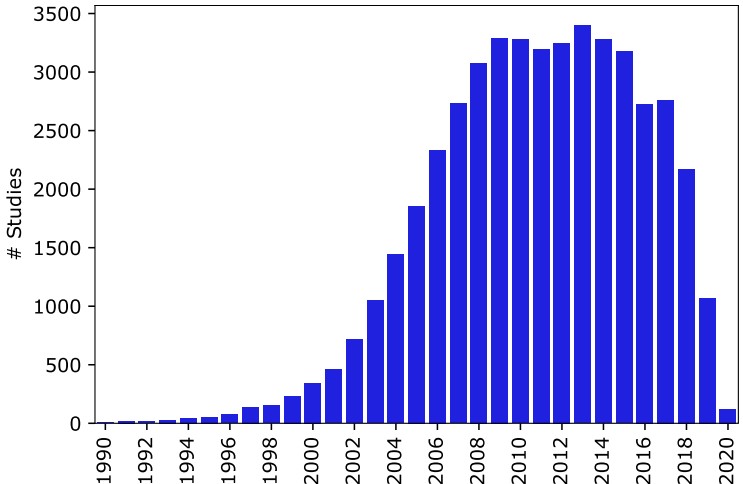

**Fig 1. Number of U.S. based clinical trials per year.**

**Candidate predictors.** We are interested in predicting the enrollment rate of clinical trials prior to the trial start; therefore, we used information available before the initiation of the trials as candidate predictors (i.e. features). All the features (Table 1) listed are derived from the *ClinicalTrials.gov* XML data, except for two: *study population* and *institution score*.

We included the Medical Subject Headings (MeSH) as features since they contain information regarding the research topic of the clinical trial. Two different classes of models were investigated: models trained using all 4,624 observed MeSH terms, and models trained on only the most frequently occurring MeSH terms. For the latter case, only MeSH terms that appear in at least 200 clinical trials were included as features. This resulted in the inclusion of 112

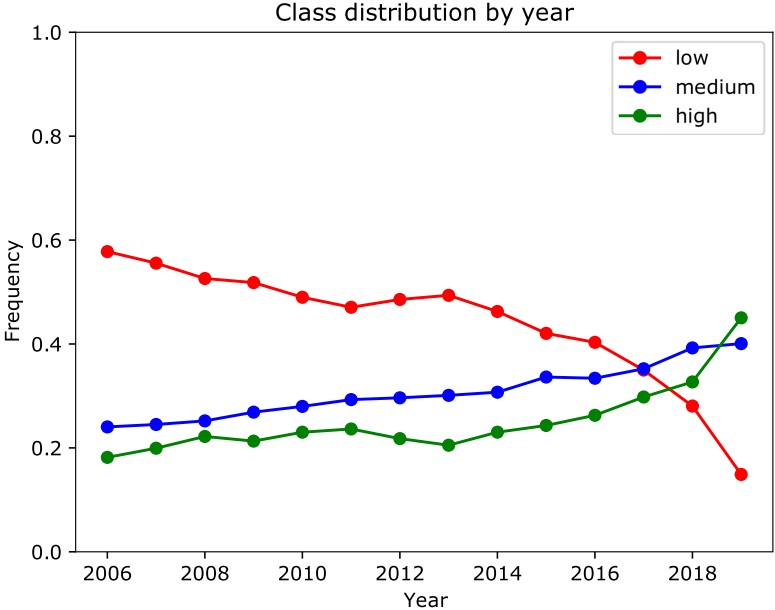

**Fig 2. Distribution of enrollment rate category over time.**

**Table 1. Characteristics of the clinical trials.** Summary statistics (percentage for categorical variables, median with interquartile range for continuous variables) of features in each enrollment rate categories were shown. "missing" level represents missing value. Only the top 5 most prevalent MeSH terms were shown.

| Clinical trials categorical features | | | |
|---|---|---|---|
| Study Phase | Enrollment Rate Class | | |
| | Low (%) | Medium (%) | High (%) |
| Early Phase 1 | 1.1 | 0.3 | 0.1 |
| Phase 1 | 9.2 | 4.4 | 4.2 |
| Phase 1/Phase 2 | 3.3 | 0.8 | 0.3 |
| Phase 2 | 11.4 | 3.9 | 3.0 |
| Phase 2/Phase 3 | 0.7 | 0.5 | 0.3 |
| Phase 3 | 1.5 | 1.9 | 3.9 |
| Phase 4 | 3.8 | 2.5 | 1.8 |
| N/A | 17.3 | 14.4 | 9.5 |
| Agency Class | Low (%) | Medium (%) | High (%) |
| Industry | 7.1 | 9.0 | 13.4 |
| NIH | 2.9 | 0.8 | 0.3 |
| Other | 36.7 | 17.8 | 8.8 |
| U.S. Fed | 1.5 | 1.1 | 0.5 |
| Randomized Allocation | Low (%) | Medium (%) | High (%) |
| N/A | 17.8 | 4.0 | 2.5 |
| Non-Randomized | 7.0 | 2.5 | 1.5 |
| Randomized | 22.8 | 22.0 | 19.0 |
| missing | 0.6 | 0.1 | 0.1 |
| Intervention Model | Low (%) | Medium (%) | High (%) |
| Crossover Assignment | 4.5 | 2.8 | 3.2 |
| Factorial Assignment | 0.6 | 0.7 | 0.6 |
| Parallel Assignment | 19.3 | 18.8 | 15.3 |
| Sequential Assignment | 0.5 | 0.3 | 0.2 |
| Single Group Assignment | 22.9 | 5.9 | 3.8 |
| missing | 0.4 | 0.1 | 0.1 |
| Intervention Type | Low (%) | Medium (%) | High (%) |
| Behavioral | 6.0 | 7.2 | 4.6 |
| Biological | 3.5 | 1.2 | 1.2 |
| Combination Product | 0.1 | 0.0 | 0.1 |
| Device | 4.5 | 3.0 | 2.5 |
| Diagnostic Test | 0.1 | 0.1 | 0.1 |
| Dietary Supplement | 1.6 | 1.1 | 0.4 |
| Drug | 25.5 | 11.3 | 11.4 |
| Genetic | 0.1 | 0.0 | 0.0 |
| Other | 4.0 | 3.4 | 2.4 |
| Procedure | 2.4 | 1.2 | 0.5 |
| Radiation | 0.6 | 0.0 | 0.0 |
| Gender Eligibility | Low (%) | Medium (%) | High (%) |
| All | 41.6 | 24.2 | 20.2 |
| Female | 4.4 | 3.3 | 2.0 |
| Male | 2.3 | 1.1 | 0.9 |
| Healthy Volunteers Eligibility | Low (%) | Medium (%) | High (%) |
| Accepts Healthy Volunteers | 9.9 | 11.3 | 11.3 |
| No | 38.3 | 17.4 | 11.8 |

(*Continued*)

**Table 1.** (Continued)

| MeSH Term | Low (%) | Medium (%) | High (%) |
|---|---|---|---|
| Diabetes Mellitus | 1.84 | 3.13 | 2.95 |
| Depression | 2.33 | 2.79 | 1.73 |
| Breast Neoplasms | 3.60 | 1.51 | 0.55 |
| Depressive Disorder | 1.96 | 2.15 | 1.50 |
| Syndrome | 2.85 | 1.25 | 0.73 |
| Clinical trials continuous features | | | |
| | Enrollment Rate Class | | |
| | Low | Medium | High |
| Eligibility Minimum Age | 18 (0) | 18 (0) | 18 (0) |
| Eligibility Maximum Age | 99 (34) | 80 (44) | 80 (44) |
| Facility Count | 1 (1) | 1 (1) | 1 (4) |
| Population (Million) | 4.7 (7.6) | 4.3 (11) | 5.9 (18) |
| Institution Score | 0 (69) | 0 (47) | 0 (17) |

MeSH terms. An occurrence count of 200 corresponds to 0.4% of the clinical trials. The most common MeSH term, *Diabetes Mellitus*, appeared 1151 times, which corresponds to 2.5% of the clinical trials. In total, 53% of the clinical trials in the data set contain at least one of the 112 frequently occurring MeSH terms.

The study population feature represents the size of the population from which participants can be recruited. For a clinical trial with a single study center, the study population is the population of the enclosing metropolitan or micropolitan area. For clinical trials with multiple study centers, the study population is the sum of population over all metropolitan and micropolitan areas that are represented by one or more study centers. Statistics on metropolitan and micropolitan area populations were obtained from the U.S. Census Bureau [21] from the url https://www2.census.gov/programs-surveys/popest/tables/2010-2018/state/totals/PEP_2018_PEPANNRES.zip.

We used the Nature Index scores for the 2018 calendar year to associate an institution score to each clinical trial to quantify the research capacity and output of the institution responsible for the clinical trial. The data can be obtained from https://www.natureindex.com/annual-tables/2018/institution/all/all/ and selecting "United States of America" as the region/country. The institution associated with a clinical trial was defined to be the institution of the principal investigator for the clinical trial. The Nature Index metrics calculate each institution's annual contribution of research articles to a curated list of 82 high impact journals. For each article, a share is assigned to each institution involved in writing an article [22]. Shares are assigned to institutions according to the proportion of the authors of an article from each institution—the total share for each article is 1.0. Harvard University had the highest 2018 Nature Index share among U.S. institutions, with a share of 875. The median share among the top 500 U.S. institutions was 7.7. The Nature Index metrics were only available for the top 500 U.S. institutions, so we assigned a share of 0 to institutions that are not included in the top 500. In theory, each record in the data set should use an institution score that was calculated before the study began, to ensure that information about future institution performance does not effect the learning procedure. However, the annual Nature Index scores were only calculated starting in 2016. We assume that institution score is a mostly time-independent metric, so that the 2018 institution score can be incorporated into the data set without creating a significant bias in the results.

### Analytical strategy

**Performance estimation and model selection.** To evaluate the performance of predicting future enrollment based on historical data and select the best model for the task, we implemented a nested time-series cross validation design [23–25]. Time series cross-validation is a modified version of cross-validation where the observations in the training data occur prior to those in the validation data. Nested cross-validation consists of two levels of cross-validation. The inner cross-validation is used to optimize the model hyper-parameter settings, and the outer cross-validation measures the performance of the hyper-parameter tuned models. In our study, both the inner and outer cross-validations were time series cross-validations. Specifically, we considered 14 validation data sets for the outer cross-validation, each corresponding to one year of data in the period 2006–2019. Data collected from years prior to the validation data were used to train and select the best performing hyper-parameters for the models via a cross-validated hyper-parameter grid search.

Since we are interested in predicting low, medium, and high enrollment rates, we considered the following multi-class classification algorithms (with multiple choices for hyper-parameters where applicable, see S1 Table in S1 File) for constructing the predictive models: multinomial logistic regression, k-nearest neighbors (KNN), multinomial elastic net, support vector machine (SVM), and random forest. The best hyper-parameter combination for each classifier was selected on the training data in the inner loop of the nested time-series cross validation. We included a dummy classifier as reference for the performance metrics that depend on class distribution. The dummy classifier makes predictions according to the prior distribution of the target variable in the training set.

**Variable scaling and missing values.** Variable scaling and treatment for missing values were incorporated into the modeling pipeline, so that scaling and imputation on the validation data are done according to the distribution of the training data. This prevents information leak from the validation data into the model training phase. The continuous variables were transformed using a standardizing scaler, except for the study population, which was scaled with a min-max scaler due to the skewed distribution for that variable. The standardizing scaler recenters a variable by subtracting the sample mean of the variable and then rescales the variable by dividing by the sample standard deviation. The min-max scaler rescales a variable to the range [−1, 1]. Median imputation was used for the continuous variables, and missing indicator columns were added to retain the missingness information. For the categorical variables with missing data, we added a "missing" level to the categories for that variable to represent the missingness information.

**Performance metrics.** We used the multi-class area under the receiver operating characteristic curve (AUC), accuracy, recall, and precision as the metrics for evaluating the predictive performance of the models. Let $c_i$ be the class labels and let $K$ be the total number of classes. Then the formula for the multi-class AUC [26, 27] is

$$\text{AUC} = \frac{2}{K(K-1)} \sum_{i=1}^{K} \sum_{\substack{j=1 \\ j \neq i}}^{K} \text{AUC}\left(c_i, c_j\right) \tag{1}$$

where $\text{AUC}(c_i, c_j)$ is the two-class AUC with $c_i$ as the positive class and $c_j$ as the negative class. Note that in the multi-class case, $\text{AUC}(c_i, c_j) \neq \text{AUC}(c_j, c_i)$.

For recall, we use the *macro-average recall*, which averages the positive identification rate for each class. The macro-average recall is invariant to changes in the class distribution [26]. For any set of predictions from a classifier, we can tabulate a *confusion matrix*, *C*, such that the $C_{ij}$ element of the confusion matrix is a count of the observations for which the predicted class

is $c_i$ and the true class is $c_j$. Let $m_j = \sum_{i=1}^{K} C_{ij}$ be the total number of samples with true class $c_j$. Then the macro-average recall is calculated as

$$\text{Recall} = \frac{1}{K} \sum_{j=1}^{K} \frac{C_{jj}}{m_j} \qquad (2)$$

In a similar fashion, the macro-average precision can be defined as

$$\text{Precision} = \frac{1}{K} \sum_{i=1}^{K} \frac{C_{ii}}{n_i} \qquad (3)$$

where $n_i = \sum_{j=1}^{K} C_{ij}$ is the total number of observations with predicted class $c_i$. Finally, the multi-class accuracy is defined analogously to the binary accuracy:

$$\text{Accuracy} = \frac{1}{N} \sum_{i=1}^{K} C_{ii} \qquad (4)$$

where $N$ is the total number of observations.

For an in-depth discussion of the properties of these metrics, please see [28, 29]

**Information content analysis.** In order to examine the predictive performance of different types of features in the data set, we trained classifiers on different sets of features and compared the predictive performances to the model trained with all features. We considered the following feature sets: (1) *population*: the population from which participants can be recruited; (2) *study center*: population, facility count, and institution score; (3) *study design*: information related to the design of the clinical trial; (4) *MeSH*: characteristics regarding the medical domain of the trial; (5) *complete*: items (1)-(4). The features belonging to each set are shown in Table 2.

Additionally, feature sets including MeSH terms (feature sets (4) and (5)) were instantiated in two ways, either using all 4,624 observed MeSH terms, or using only the most frequently observed MeSH terms. Except where otherwise noted, results below are reported for models trained using all MeSH terms.

**Selection bias and domain adaptation.** Only completed clinical trials were included in this study, since the total enrollment is not known until a trial is completed. In the more recent

**Table 2. The feature sets used for information content analysis.**

| Feature | Complete | Population | Study Center | Study Design | MeSH |
|---|---|---|---|---|---|
| Study phase | X | | | X | |
| Funding agency class | X | | | X | |
| Randomized allocation | X | | | X | |
| Intervention model | X | | | X | |
| Intervention type | X | | | X | |
| Eligibility gender | X | | | X | |
| Accepts healthy volunteers | X | | | X | |
| Eligibility minimum age | X | | | X | |
| Eligibility maximum age | X | | | X | |
| Study center count | X | | X | | |
| Study population | X | X | X | | |
| Institution score | X | | X | | |
| MeSH terms | X | | | | X |

years, the number of completed trials shows a decreasing trend (Fig 1). This is due to the fact that many of the trials that were started in recent years have not completed yet. This introduces a selection bias by including only shorter duration studies in the years close to 2020 (S1, S3 Figs in S1 File) and introduces a shift in the distribution of the enrollment rate (S2 Fig in S1 File) and enrollment rate categories (Fig 2).

As a result, a difference in the distribution is present between the training data (earlier in time) and the validation data (more recent in time), especially for the more recent years. The distribution shift may impact model performance, since most machine learning techniques assume that training and validation data are drawn from the same distribution. Therefore, we implemented domain adaptation, a family of techniques designed to alleviate the effects of distribution shift [30–33]. Specifically, We employed an approach where training samples are weighted according to their likelihood in the validation set.

Based on the chronological nature of the study duration selection bias in our experiment, we derived the following simple formula for the weights:

$$w_i = I(t_i < \tau) \tag{5}$$

where $w_i$ is the weight for training sample $i$, $t_i$ is the study duration for sample $i$, and $\tau$ is the maximum possible study duration that can occur in the validation set. $I$ is the indicator function. The resulting weight for each sample is either 0 or 1. Therefore, applying these weights to the training data is equivalent to filtering out the training samples with duration longer than $\tau$. A mathematical justification for this approach can be found in Supplemental Information on mathematical justification for domain adaptation. We trained models with and without domain adaptation to investigate the impact of the selection bias.

## Results

### Characteristics of the *ClinicalTrials.gov* data

We analyzed 46,724 U.S. based clinical trials from 1990–2020. The distribution of the enrollment rate categories over time is shown in Fig 2. The class distribution for enrollment rate showed a shift over time, where the proportion of clinical trials with low enrollment rates decreased over time and the proportion of clinical trials with medium and high enrollment rates increased. This is partly due to the fact that we only considered clinical trials completed by October of 2020. Therefore, the more recent clinical trials included in the analysis have shorter study duration. There is an inverse relationship between study duration and enrollment rate S3 Fig in S1 File, so the bias towards shorter clinical trials results in a bias towards higher enrollment rates.

We used information extracted from *ClinicalTrials.gov* and external sources (population count of the location of the recruitment and institutional score) as features for enrollment rate. The summary statistics of these features are included in Table 1.

### Predictive performance of models built with all features

A comparison of the performance on the validation set (years 2006 to 2019) of the models constructed by various classification methods using all 4,636 candidate predictors is shown in Fig 3. The predictive performance of all evaluated classification methods outperformed the dummy classifier, where predictions were made according to the distribution of the target variable in the training data. This indicates that the features, which capture information available prior to the initiation of the trial, contain information regarding the enrollment rate. Furthermore, the predictive performance of the models constructed by the various classification

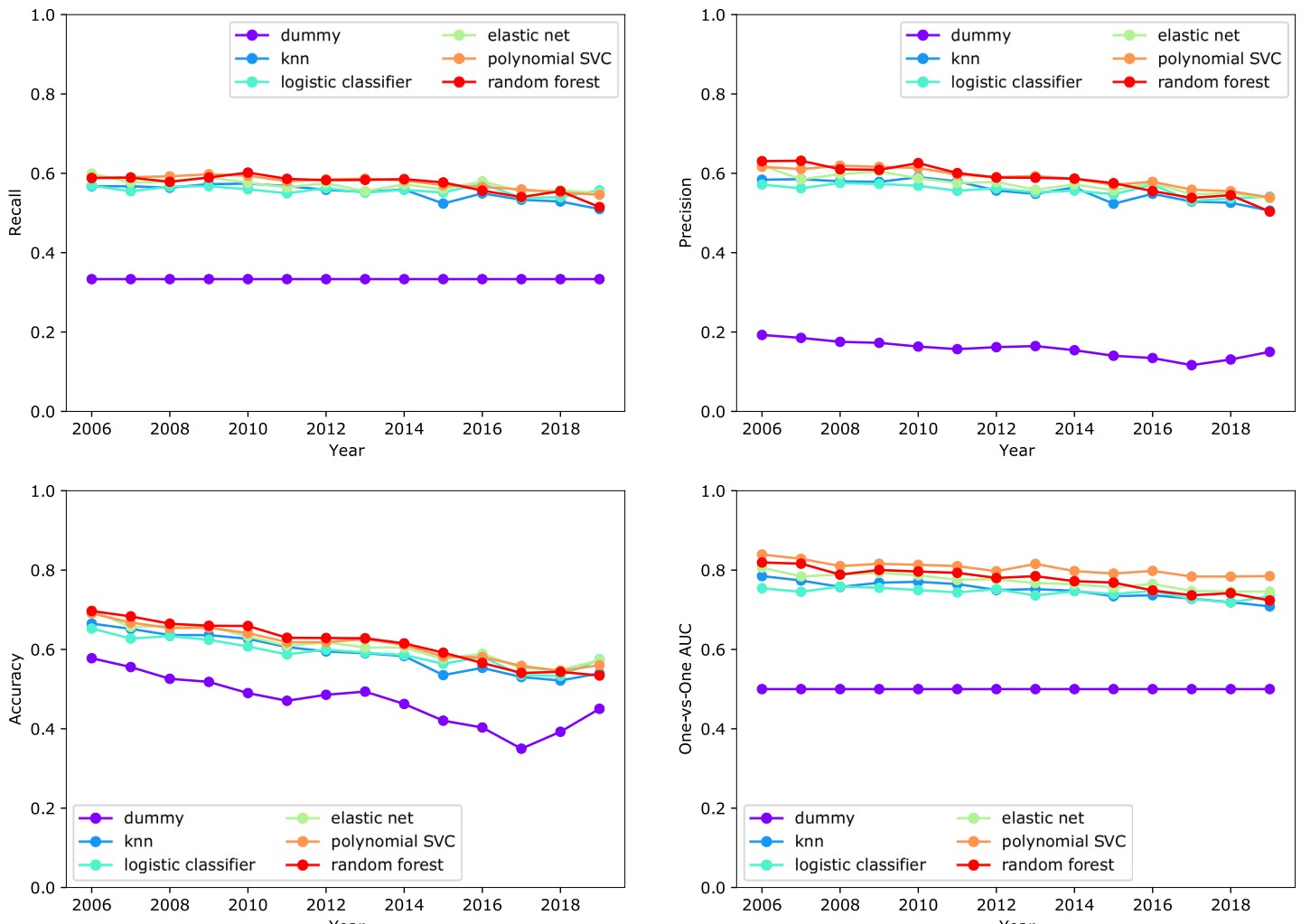

**Fig 3. Predictive performance of models constructed with various classification methods using the complete set of 4,636 features.** The predictive performance for the various classifiers is similar, and each outperforms the dummy classifier. Note: On most plots, the performance of the logistic classifier is not visible, since its performance is the same as the elastic net.

methods was very similar across all the performance metrics. The SVM resulted in the best overall performance over years 2006 to 2019 with AUC = 0.81 ± 0.02 (*mean ± std*), recall = 0.58 ± 0.02, precision = 0.59 ± 0.04, and accuracy = 0.62 ± 0.05. The random forest model was the second best and only performed minimally worse compared to the SVM with AUC = 0.78 ± 0.03 (*mean ± std*), recall = 0.57 ± 0.02, precision = 0.58 ± 0.04, and accuracy = 0.62 ± 0.05. The performance of the other classifiers is show in Fig 3 and also listed in S2 Table in S1 File.

We also investigated the model performance on a reduced variable set where MeSH terms that appeared less than 200 times were not included. For the feature set containing these most frequent MeSH terms along with all other features. The SVM classifier did not converge for experimental setting in the allotted run time of 96 hours. Among the methods that finished execution in all experimental setting. The best model was the random forest classifier. The random forest had similar performance compared to when all MeSH terms were used, with average performance over years 2006 to 2019 of AUC = 0.78 ± 0.02 (*mean ± std*),

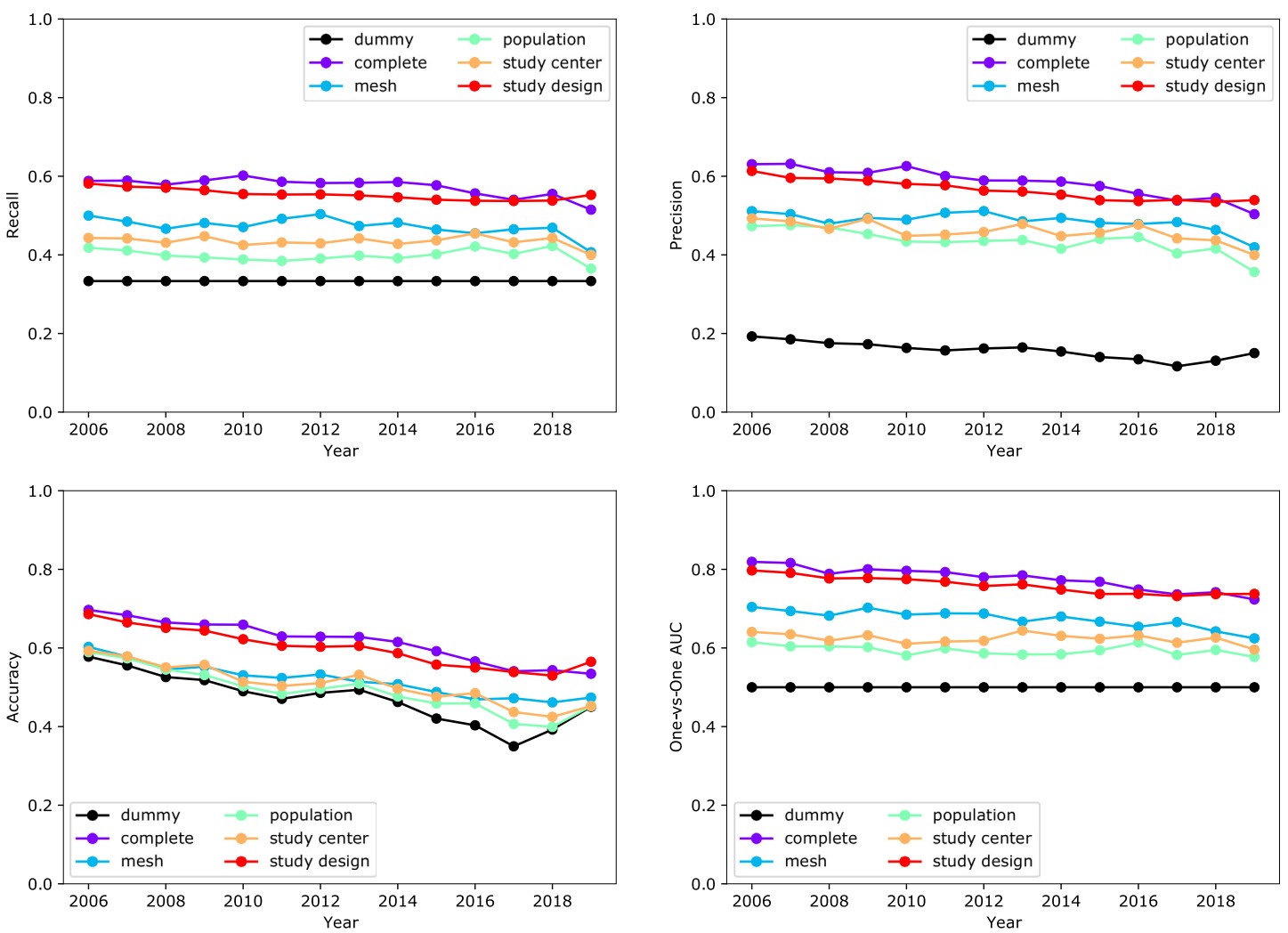

**Fig 4. Predictive performance of random forest models with different feature subsets.**

recall = 0.57 ± 0.01, precision = 0.58 ± 0.03, and accuracy = 0.62 ± 0.04. The performance of the other classifiers is listed in S3 Table in S1 File.

## Information content in different types of features

To assess the relative information content in various feature types, we built models using feature sets listed in Table 2. We report below the results from the random forest classifier (see Fig 4), since SVM, the best overall classifier on the complete feature set, failed to converge on two feature types of interest. The results from all classifiers were similar and can be found in S2 Table in S1 File. The feature set containing study design related information (e.g. phase, funding agency, randomization, intervention, and eligibility criterion) resulted in the best predictive performance among all the feature sets, with an AUC = 0.76 ± 0.02 over all years. All other feature sets examined contained weak to moderate predictive information regarding enrollment rate on their own. The models based on only the population of the recruitment region, the study center characteristics, and the MeSH terms resulted in AUC of 0.59 ± 0.01, 0.62 ± 0.01, and 0.67 ± 0.02, respectively. Moreover, adding those features to the study design

features only provided a marginal increase in model performance—the AUC of the model with the complete set of features was 0.78 ± 0.03, while the AUC of the model with the study design features was 0.76 ± 0.02.

## Selection bias and domain adaptation

Due to the selection bias present in the more recent years of data (e.g. Fig 2), we employed domain adaptation and compared the predictive performance of models with and without domain adaptation. The effect of the domain adaptation on model performance was negligible when models were trained on the full data set, including all MeSH terms.

However, domain adaptation had a larger impact when models were trained on the data set where MeSH terms that appeared less than 200 times were not included. This result is demonstrated in Fig 5 for a random forest model, where it is seen that domain adaptation brought the

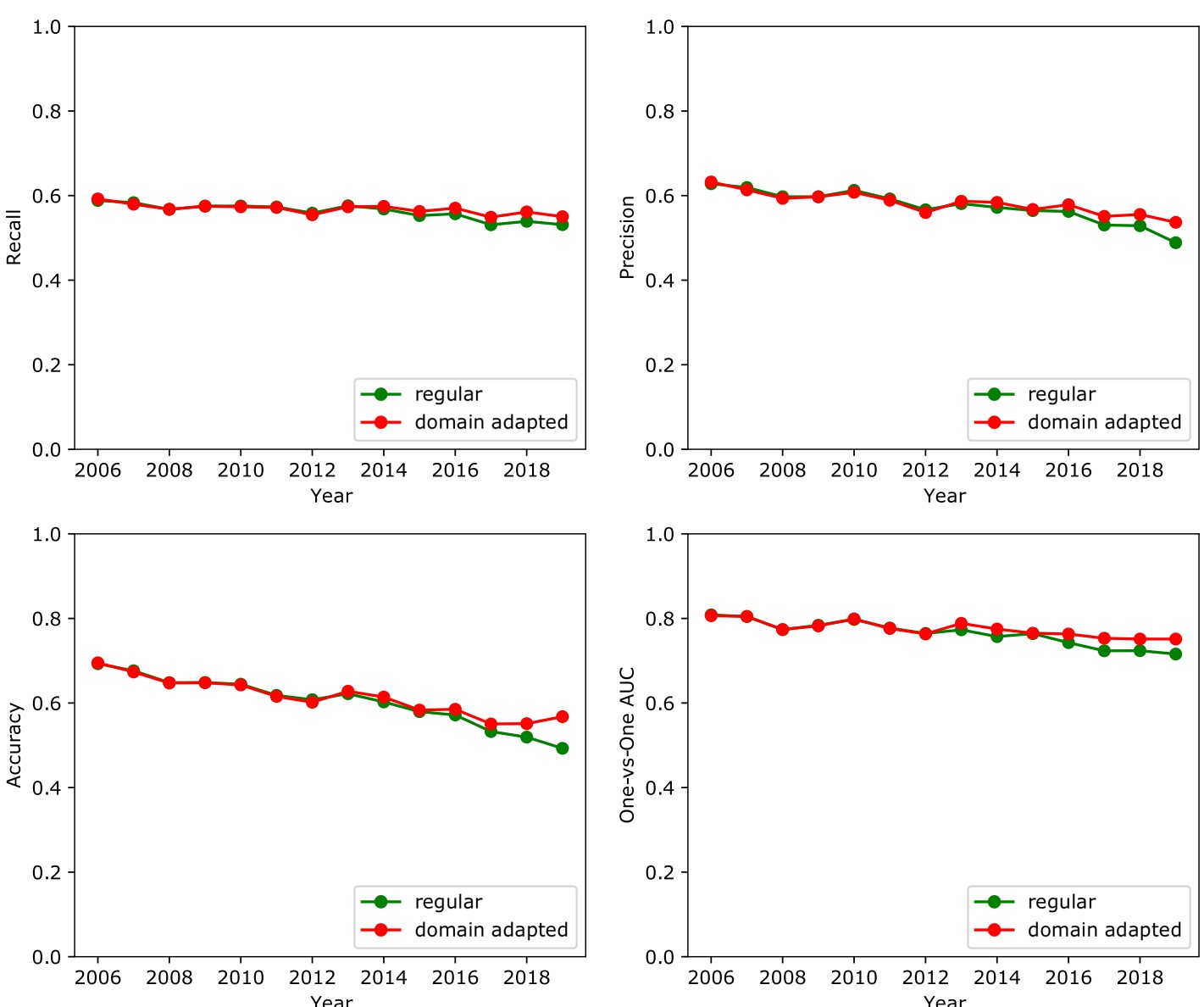

**Fig 5. Predictive performance of random forest models with and without domain adaptation on dataset with reduced MeSH features.**

performance for recent years more in line with the performance for the earlier years. The average AUC across 2016 to 2019 was $0.76 \pm 0.01$ and $0.73 \pm 0.01$ with and without domain adaptation. The impact of domain adaptation on the predictive performance for the earlier years was minimal, since for those years the distribution shift between the training and validation data was absent or minimal.

## Discussion

### Prediction of clinical trials enrollment rate

In the current study, we quantified the predictive signal for clinical trial enrollment rate in trial characteristics available prior to their initiation. We adopted a nested time-series cross-validation design. This design takes into account the data availability with respect to time for both model selection and performance estimation, reducing the bias for estimating future model performance. We also explored a variety of predictive modeling supervise learning methods. Our modeling procedure achieved good predictive performance that is stable over time and across different classifiers. This indicates that the modeling procedure produced models that were stable and generalizable to future data. Moreover, using a reduced feature set where MeSH terms that appeared less than 200 times were excluded resulted in similar performance. With respect to the information content in various types of features examined, we found that the features regarding study design (including study phase, funding agency, randomization, interventions, and eligibility) were the most informative compared to other feature subsets. The other feature subsets (population, study center, and MeSH terms) contain information regarding enrollment rate, but the information is largely overlapping with the information in the study design related features, since the combined information in all features only marginally improves upon the information in the study design features.

Our study revealed that the *ClinicalTrials.gov* data contains rich information that can be leveraged for predicting enrollment rate. We also showed that external information (population of recruitment location and information regarding the recruiting institution) can be retrieved to supplement data collected by *ClinicalTrials.gov*.

### Limitations and future work

We only examined structured field from the *ClinicalTrials.gov* as features for predicting enrollment rate. Unstructured free-text data summarizing the goal and procedures of each study is also available from the *ClinicalTrials.gov*. Future work could employ Natural Language Processing methods to extract information and construct features from the free-text data for enrollment rate prediction. Natural Language Processing methods applied to the free-text data from *ClinicalTrials.gov* has been explored for various applications including construction of knowledge base for eligibility criterion [34], illustration of relationships among multiple clinical trials [35], and literature mining for trends and prevalence of clinical instruments [36]. However, using Natural Language Processing methods for feature construction for predictive task using the *ClinicalTrials.gov* data is an untapped area. We expect the combination of free-text based features and structured data could improve the predictive performance for enrollment rate, as demonstrated in prior studies in other data domain [37–39].

Also, The current study limited the external data source to the study population obtained from the U.S. census Bureau [21] and the institutional score defined by the Nature Index score [22]. For future work, we believe exploring additional data sources for enrollment rate may further improve predictive performance for enrollment rate. Examples from a variety of problem domains has demonstrated that obtaining information from multiple sources and integrating them with machine learning methods can result in high quality models [40–43].

Potential external sources of information include information related to the principle investigator of the trial ([40] showed several possible feature constructions related to this), the prevalence of the disease or condition in question near the recruitment location ([44] demonstrated a way to construct these variables), and the characteristics of the population (age, gender, education, insurance status, etc., can be obtained from census data [21]) near the recruitment location.

## Conclusions

The current study emperically demonstrated the feasibility of enrollment prediction prior to the initiation of clinical trials. To the best of our knowledge, this is the first study that employs machine learning methods for enrollment rate prediction based on a large and comprehensive sample of U.S.-based clinical trials. This study is the first step towards a data-driven decision support system for assessing whether a proposed clinical trial would likely meet its enrollment goal.

## Supporting information

**S1 File.**
(PDF)

## Acknowledgments

The authors thank Dr. David Haynes II for his advice regarding the census data. The authors thank the Minnesota Supercomputing institute for providing the high performance computing resource.

## Author Contributions

**Conceptualization:** Constantin Aliferis, Sisi Ma.

**Data curation:** Cameron Bieganek, Sisi Ma.

**Formal analysis:** Cameron Bieganek.

**Funding acquisition:** Sisi Ma.

**Investigation:** Cameron Bieganek, Constantin Aliferis, Sisi Ma.

**Methodology:** Cameron Bieganek, Constantin Aliferis, Sisi Ma.

**Project administration:** Sisi Ma.

**Resources:** Constantin Aliferis, Sisi Ma.

**Software:** Cameron Bieganek.

**Supervision:** Constantin Aliferis, Sisi Ma.

**Validation:** Cameron Bieganek.

**Visualization:** Cameron Bieganek.

**Writing – original draft:** Cameron Bieganek, Sisi Ma.

**Writing – review & editing:** Cameron Bieganek, Constantin Aliferis, Sisi Ma.

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
