## [Decision Letter · Decision Letter 0]

20 Dec 2021

PONE-D-21-37339Prediction of clinical trial enrollment ratesPLOS ONE

Dear Dr. Ma,

Thank you for submitting your manuscript to PLOS ONE. After careful consideration, we feel that it has merit but does not fully meet PLOS ONE’s publication criteria as it currently stands. Therefore, we invite you to submit a revised version of the manuscript that addresses the points raised during the review process.

We look forward to receiving your revised manuscript.

Kind regards,

Sathishkumar V E

Academic Editor

PLOS ONE

Journal Requirements:

"SM’s time on this work is partially supported by Grant UL1TR002494"

"SM's time on this work is partially supported by Grant UL1TR002494. "

Reviewers' comments:

Reviewer's Responses to Questions

**Comments to the Author**

1. Is the manuscript technically sound, and do the data support the conclusions?

Reviewer #1: Yes

Reviewer #2: Yes

2. Has the statistical analysis been performed appropriately and rigorously? 

Reviewer #1: Yes

Reviewer #2: Yes

3. Have the authors made all data underlying the findings in their manuscript fully available?

Reviewer #1: Yes

Reviewer #2: No

4. Is the manuscript presented in an intelligible fashion and written in standard English?

Reviewer #1: No

Reviewer #2: Yes

5. Review Comments to the Author

Reviewer #1: The Research Paper needs the following Major Revisions and is Subject for re-review, and after re-review, the final decision for the paper will be done:

a. Abstract- In the last lines, highlight regarding experimental analysis.

b. Introduction should be more broad and should cover more towards Problem Definition and Scope.

Add Objectives of the paper at the end of Introduction

Add Organization of the paper

c. Add some Literature review to this paper with min 4-10 Papers.

d. Add some case study based discussion to the paper.

e. Add conclusion and future scope to the paper

f. Addition of min 5-10 Latest references of 2021 cum 2022 are recommended to th epaper.

Reviewer #2: This article performs the clinical examination on 4,636 candidate predictors based on data collected by ClinicalTrials.gov and external sources for enrollment rate prediction using various state-of-the-art machine learning methods.This article studied well and may be considered for publication after addressing the below:

1. Highlight the contributions in the Introduction.

2. Provide the numbering for subsections of the article.

3. PRovide the literature study.

4. The formulae available in the existing papers must be cited.

5. Add the conclusion to the article.

6. List the limitations for the proposed work.

7. How the authors considered missing values in the datasets?

6. PLOS authors have the option to publish the peer review history of their article (what does this mean?). If published, this will include your full peer review and any attached files.

Reviewer #1: **Yes: **Anand Nayyar

Reviewer #2: No

---

## [Author Response · Author response to Decision Letter 0]

10 Jan 2022

The response to review is attached with the revised manuscript (at the end of the combined pdf).

---

## [Decision Letter · Decision Letter 1]

14 Jan 2022

Prediction of clinical trial enrollment rates

PONE-D-21-37339R1

Dear Dr. Ma,

We’re pleased to inform you that your manuscript has been judged scientifically suitable for publication and will be formally accepted for publication once it meets all outstanding technical requirements.

Kind regards,

Sathishkumar V E

Academic Editor

PLOS ONE

Additional Editor Comments (optional):

Reviewers' comments:

Reviewer's Responses to Questions

**Comments to the Author**

1. If the authors have adequately addressed your comments raised in a previous round of review and you feel that this manuscript is now acceptable for publication, you may indicate that here to bypass the “Comments to the Author” section, enter your conflict of interest statement in the “Confidential to Editor” section, and submit your "Accept" recommendation.

Reviewer #1: All comments have been addressed

Reviewer #2: All comments have been addressed

2. Is the manuscript technically sound, and do the data support the conclusions?

Reviewer #1: Yes

Reviewer #2: Yes

3. Has the statistical analysis been performed appropriately and rigorously? 

Reviewer #1: Yes

Reviewer #2: Yes

4. Have the authors made all data underlying the findings in their manuscript fully available?

Reviewer #1: Yes

Reviewer #2: Yes

5. Is the manuscript presented in an intelligible fashion and written in standard English?

Reviewer #1: Yes

Reviewer #2: Yes

6. Review Comments to the Author

Reviewer #1: The Paper stands Accepted with no further revisions. All the comments are addressed and suitable revisions are done.

Reviewer #2: The authors addressed all the comments and the current version of the article is recommended for the publication.

7. PLOS authors have the option to publish the peer review history of their article (what does this mean?). If published, this will include your full peer review and any attached files.

Reviewer #1: **Yes: **Anand Nayyar

Reviewer #2: No

---

## [Editor Report · Acceptance letter]

10 Feb 2022

PONE-D-21-37339R1 

Prediction of clinical trial enrollment rates 

Dear Dr. Ma:

I'm pleased to inform you that your manuscript has been deemed suitable for publication in PLOS ONE. Congratulations! Your manuscript is now with our production department. 

Kind regards, 

on behalf of

Dr. Sathishkumar V E 

Academic Editor

PLOS ONE